# Decay and Termite Resistance of Wood Modified by High-Temperature Vapour-Phase Acetylation (HTVPA), a Simultaneous Acetylation and Heat Treatment Modification Process

**DOI:** 10.3390/polym16111601

**Published:** 2024-06-05

**Authors:** Zhong-Yao Wang, Jin-Wei Xu, Jian-Wei Liu, Ke-Chang Hung, Tung-Lin Wu, Wen-Shao Chang, Jyh-Horng Wu

**Affiliations:** 1Department of Forestry, National Chung Hsing University, Taichung 402, Taiwan; hi4u2ca707@gmail.com (Z.-Y.W.); ecsgunro@gmail.com (J.-W.X.); ted2301636@gmail.com (J.-W.L.); 2Department of Wood Based Materials and Design, National Chiayi University, Chiayi 600, Taiwan; hkc@mail.ncyu.edu.tw; 3Department of Wood Science and Design, National Pingtung University of Science and Technology, Pingtung 912, Taiwan; tony.npust@gmail.com; 4Lincoln School of Architecture and the Built Environment, University of Lincoln, Lincoln LN6 7TS, UK; wchang@lincoln.ac.uk; 5Advanced Plant and Food Crop Biotechnology Center, National Chung Hsing University, Taichung 402, Taiwan

**Keywords:** Japanese cedar, dimensional stability, biological durability, high-temperature vapour-phase acetylation, weight percent gain

## Abstract

High-temperature vapour-phase acetylation (HTVPA) is a simultaneous acetylation and heat treatment process for wood modification. This study was the first investigation into the impact of HTVPA treatment on the resistance of wood to biological degradation. In the termite resistance test, untreated wood exhibited a mass loss (ML_t_) of 20.3%, while HTVPA-modified wood showed a reduced ML_t_ of 6.6–3.2%, which decreased with an increase in weight percent gain (WPG), and the termite mortality reached 95–100%. Furthermore, after a 12-week decay resistance test against brown-rot fungi (*Laetiporus sulfureus* and *Fomitopsis pinicola*), untreated wood exhibited mass loss (ML_d_) values of 39.6% and 54.5%, respectively, while HTVPA-modified wood exhibited ML_d_ values of 0.2–0.9% and −0.2–0.3%, respectively, with no significant influence from WPG. Similar results were observed in decay resistance tests against white-rot fungi (*Lenzites betulina* and *Trametes versicolor*). The results of this study demonstrated that HTVPA treatment not only effectively enhanced the decay resistance of wood but also offered superior enhancement relative to separate heat treatment or acetylation processes. In addition, all the HTVPA-modified wood specimens prepared in this study met the requirements of the CNS 6717 wood preservative standard, with an ML_d_ of less than 3% for decay-resistant materials.

## 1. Introduction

To date, the Earth’s average temperature has risen by approximately 1.1 °C since the late 19th century, with emissions continuing to escalate. To align with the Paris Agreement’s requirement of limiting global warming to 1.5 °C, a reduction in emissions by 45% by 2030 and achieving carbon neutrality by 2050 are imperative. Consequently, carbon reduction efforts have become a global social responsibility investment (SRI). A key focus of this goal is natural carbon sinks, which refer to natural environments that capable of capturing and storing carbon dioxide, such as forest, soil, and marine ecosystems. These sinks effectively capture carbon dioxide from the atmosphere, helping to mitigate climate change. Utilising wood not only enhances carbon sequestration in forests but also aligns with the principles of a circular economy, contributing to the goal of net-zero carbon emissions. Japanese cedar (*Cryptomeria japonica* (L.f.) D. Don) is a prominent plantation tree species in East Asia [1,2]. Its extract exhibits excellent biological activity and can be used as a natural insecticide, wood preservative, fungicide, as well as in health care and medical products [3,4,5,6,7,8]. It is also the highest-yielding domestic timber in Taiwan; however, its advanced age leads to inferior properties. Recognising these attributes, this study selected Japanese cedar wood as the experimental material to explore property improvements, particularly in resistance to biological degradation.

In general, wood exhibits moisture sensitivity, anisotropy, biodegradability, and susceptibility to photodamage. Therefore, when used outdoors, wood is susceptible to environmental factors, such as sunlight, moisture, and biological agents, which can lead to defects, such as cracking, deformation, decay, erosion, and discolouration, limiting its utility and lifespan [9]. To extend the service life of wood, many researchers have developed wood modification methods that have been proven to enhance its properties, such as heat treatment, acetylation, and inorganic modification using so-gel technology. Of these methods, wood acetylation is one of the most widely applied and currently commercialised wood modification techniques [10,11]. Wood acetylation is a nontoxic and environmentally friendly modification method that involves an esterification reaction using the available hydroxyl groups in wood with acetic anhydride (AA) or other chemicals (e.g., acetic acid, vinyl acetate, and acetyl chloride, etc.) to form acetyl groups. This reaction reduces the hydrophilic hydroxyl content in the wood, thus lowering its hydrophilicity [12]. Moreover, acetylation has been proven to improve the dimensional stability, fungal decay resistance, termite resistance, and weathering resistance of wood [13,14,15,16].

Conventional liquid-phase acetylation processes involve immersing the wood in chemical agents and conducting the reactions through heat conduction. These processes are associated with high chemical consumption, potential energy waste, environmental impacts, and an increase in the overall weight of the wood, which in turn increases transportation costs. Recent studies have found that vapour-phase acetylation treatment can produce acetylated wood with a modified gradient and effectively reduce the consumption of modification chemicals [17,18]. Conversely, physical heat treatment techniques have gained increasing attention in the past decade since they can effectively avoid the hazards posed by chemical agents to human health and the environment. Studies have indicated that wood subjected to heat treatment not only reduces the number of hydrophilic functional groups in the wood [19], but also enhances its dimensional stability [20,21,22] and decay resistance [23,24,25]. Therefore, heat treatment is considered environmentally friendly for wood modification. However, wood modification is mostly conducted using a single method, either physical or chemical, with very few instances of simultaneously applying both approaches for wood modification. In our previous research, we have investigated the effects of high-temperature vapour-phase acetylation (HTVPA), which is a simultaneous acetylation and heat treatment modification process, on the physicomechanical properties of Japanese cedar wood, conducted at temperatures ranging from 145 to 220 °C for durations of 2 to 16 h [26]. Accordingly, we found that the optimal HTVPA treatment condition is at 200 °C for 8 h. However, the resistance of the HTVPA-modified wood to biological degradation is currently unclear. To address this research gap, the aim of this study is to investigate the decay and termite resistance of Japanese cedar wood modified by HTVPA.

## 2. Materials and Methods

### 2.1. Materials

Japanese cedar (*Cryptomeria japonica* (L.f.) D. Don), aged 40–45 years and the most common domestically produced wood in Taiwan, was supplied by Jang Chang Lumber Industry Co., Ltd. (Hsinchu, Taiwan). Flat-sawn timbers with dimensions of 500 mm × 136 mm × 25 mm and an average density of 396 kg/m^3^ were prepared from conventionally kiln-dried wood with an average moisture content of 14%. To mitigate the impacts of the biological variations among specimens, we adhered to CNS 14630 [27], which is a standard for coniferous structural timber, and conducted mechanical grading on the specimens. During the four-point bending tests, a span-to-depth ratio of 18 and a loading rate of 5 mm/min were maintained. After recording the load and deformation values at 3000 N, the modulus of elasticity (MOE) was calculated, enabling the selection of specimens that conformed to the range of E70 (5.9 ≤ MOE < 7.8 GPa). These selected specimens were further cut into dimensions of 10 mm (longitudinal) × 20 mm (radial) × 20 mm (tangential). All specimens were free of any defects, such as knots, cracks, shakes, splits, and grain deviations ensuring their suitability for the subsequent HTVPA treatment. Potato dextrose agar and analytical grade acetic anhydride (AA) were purchased from Merck (Darmstadt, Germany) and Easchem Co., Ltd. (Taipei, Taiwan), respectively.

### 2.2. High-Temperature Vapour-Phase Acetylation (HTVPA) Treatment

A semi-industrial level treatment chamber with controlled temperature and pressure (San Neng Ltd., Chiayi, Taiwan) was used for the high-temperature vapour-phase acetylation (HTVPA) treatment, as shown in Figure 1A. The selection of treatment conditions was based on findings from our previous research [26]. To obtain an accurate weight percent gain (WPG), the specimens were first placed in the treatment chamber and dried at 105 °C for 48 h, and their oven-dry weights were recorded. Subsequently, Japanese cedar specimens and AA were added to the chamber with a solid–liquid ratio of 2:1 (*w*/*w*). The specimens (approximately 17,000 cm^3^) were elevated on wooden racks and spaced at regular intervals within the chamber to ensure direct contact between the wood and the vapour-phase AA. Additionally, the AA was placed on an iron tray at the bottom of the treatment chamber to minimise the risk of splashing onto the specimens. Then, a vacuum pump was used to reduce the pressure inside the chamber to 25 kPa, and the temperature was increased to 200 °C at a rate of 2 °C/min for an 8 h HTVPA treatment. During the experiment, as shown in Figure 1B, temperature profiles were recorded using a thermocouple logger (CHY506A, Taipei, Taiwan). After the reaction was completed, the chamber was allowed to cool to room temperature, and any remaining chemicals were removed. The modified specimens were further dried at 105 °C for 24 h, and their oven-dried weights were recorded to calculate the WPG. Finally, all specimens were conditioned at 20 °C and 65% relative humidity (RH) for subsequent testing.

### 2.3. Measurement of Specimen Dimension Changes after HTVPA Treatment

Before HTVPA treatment, specimens with dimensions of 10 mm (longitudinal) × 20 mm (radial) × 20 mm (tangential) were dried in a treatment chamber at 105 °C for 48 h, and their tangential and radial dimensions were measured with seven repetitions for each group of specimens. Subsequently, after the HTVPA treatment was completed, the specimens were dried at 105 °C for 24 h, and their dimensions were measured again. The dimension change (DC) was calculated using Equation (1):(1)$$\mathrm{DC}\;(\%)=\;\frac{L_{1}-{\;L}_{0}}{L_{0}}\;\times \;100\;$$
where L_0_ and L_1_ are the oven-dry tangential or radial dimensions (mm) of the specimen before and after HTVPA treatment, respectively.

### 2.4. Density, Equilibrium Moisture Content (EMC), and Moisture Excluding Efficiency (MEE)

The EMC values of the specimens were measured following the CNS 452 [28] standard. Specimens measuring 10 mm × 20 mm × 20 mm were conditioned in an environment with a temperature of 20 °C and 65% RH until they reached constant weights. The masses of the specimens after conditioning were measured. Subsequently, the specimens were dried in an oven at 105 °C until they reached constant weight, and their oven-dry mass and dimensions were recorded. The EMC, MEE, and density values of the specimens were then calculated using Equations (2), (3), and (4), respectively, with seven replicates for each group of specimens.
(2)$$\mathrm{EMC}\;\left(\%\right)=\frac{m_{t}-{\;m}_{0}}{m_{0}}\;\times \;100\;$$
(3)$$\mathrm{MEE}\;\left(\%\right)=\;\frac{{\mathrm{EMC}}_{u}-{\mathrm{EMC}}_{h}}{{\mathrm{EMC}}_{u}}\;\times \;100$$
(4)$$\mathrm{Density}\;\left(g/{\mathrm{cm}}^{3}\right)=\;\frac{m_{0}}{V_{0}}$$
where m_0_ represents the oven-dry mass (g) of the specimen, m_t_ represents the mass of the specimen after conditioning (g), EMC_u_ represents the EMC of untreated specimens, EMC_h_ represents the EMC of HTVPA-modified specimens, V_0_ represents the oven-dry volume (cm^3^).

### 2.5. Volumetric Swelling and Anti-Swelling Efficiency (ASE)

In this test, following the standard method specified in CNS 14927 [29] for the determination of wood volumetric swelling, specimens measuring 10 mm × 20 mm× 20 mm were conditioned at 20 °C and 65% RH until they reached a constant weight. Tangential and radial dimensions were measured with seven replicates for each group of specimens. Subsequently, the specimens were dried in an oven at 105 °C, and their dimensions were measured under oven-dry conditions. Then, the specimens were soaked in distilled water at 20 ± 5 °C for one week, and their tangential and radial dimensions were measured again. This cycle of drying and soaking was repeated five times, with water soaking and oven drying, and then the volumetric swelling coefficient (S_65_) and total volumetric swelling coefficient (S_max_) values of the specimens were calculated using Equations (5) and (6). Furthermore, following the method outlined by Islam et al. [30], the anti-swelling efficiency (ASE) of the HTVPA-modified specimens was calculated as follows (Equation (7)) to assess their dimensional stability:(5)$$S_{65}\;(\%)=\frac{{(L}_{t65}\;\times \;L_{r65})-{(L}_{t0}{\;\times \;L}_{r0})}{L_{t0}{\;\times \;L}_{r0}}\;\times \;100$$
(6)$$S_{\max}\;(\%)=\frac{\left(L_{\mathrm{tmax}}{\;\times \;L}_{\mathrm{rmax}}\right)-\left(L_{t0}{\;\times \;L}_{r0}\right)}{L_{t0}{\;\times \;L}_{r0}}\;\times \;100$$
(7)$$\mathrm{ASE}\;\left(\%\right)=\frac{S_{u}-{\;S}_{h}}{S_{u}}\;\times \;100$$
where L_t65_ and L_r65_ represent the tangential and radial dimensions (mm) of the specimens after conditioning; L_tmax_ and L_rmax_ represent the tangential and radial dimensions (mm) of the specimens above the fibre saturation point; L_t0_ and L_r0_ represent the tangential and radial dimensions (mm) of the specimens in an oven-dry state; S_u_ represents the S_65_ or S_max_ of untreated specimens; and S_h_ represents the S_65_ or S_max_ of HTVPA-modified specimens.

### 2.6. Termite Resistance Test

This test followed the standard method specified in CNS 15756 [31] for wood termite resistance testing. The specimen dimensions were 10 mm × 20 mm × 20 mm, and the test termites, *Coptotermes formosanus* Shiraki, were collected from the campus of National Chung Hsing University. Prior to the test, 100 g of oven-dried construction sand was evenly distributed into a cylindrical glass container with a diameter of 75 mm and a height of 130 mm. The container was sterilised using an autoclave (TM-329, New Taipei, Taiwan) at 121 °C and 118 kPa to serve as the feeding container. Specimens were pre-dried in an oven at 60 °C for 48 h, and their masses were measured before the test. During the test, 20 g of deionised water was added to the container, and a plastic net of 30 mm × 30 mm × 1 mm was placed on top of the construction sand. The specimens were then placed in the container with the cross-section facing upwards. Subsequently, 150 worker termites and 15 soldier termites were introduced into the container, and the feeding container was placed in an environment with a temperature of 28 °C and 80% RH for 21 d. After the test, the number of deceased worker termites was tallied, and the specimens were placed in an oven at 60 °C for 48 h to measure their masses. The mass loss (ML_t_) and the termite mortality rate (TM) values of the specimens were calculated with three replicates for each group after the termite resistance test.

### 2.7. Decay Resistance Test

This test followed the performance standards for wood preservatives specified in CNS 6717 [32] to conduct the decay resistance test. The specimen dimensions were 10 mm (L) × 20 mm (W) × 20 mm (H), and the test fungi were obtained from the Bioresource Collection and Research Center (BCRC) (Hsinchu, Taiwan). The test fungi included two white-rot fungi (*Lenzites betulina* (LB) (BCRC 35296) and *Trametes versicolor* (TV) (BCRC 35253)) and two brown-rot fungi (*Laetiporus sulfureus* (LS) (BCRC 35305) and *Fomitopsis pinicola* (FP) (BCRC 35257)). Cylindrical glass containers with diameters of 95 mm and heights of 70 mm were used as culture containers. The culture medium consisted of 39 g of agar and 1 L of distilled water. Prior to the test, 30 mL of the culture medium was added to each culture container, and they were sterilised using an autoclave (121 °C, 118 kPa). The fungal strains were transferred to the centre of the culture medium, and the culture containers were sealed with paraffin wax. The strains were then placed in a growth chamber at 25 °C and 70% RH until the fungal mycelium covered the surface of the culture medium. Before decay testing, the specimens were dried in an oven at 60 °C for 48 h, and their masses were measured. Three sterilised specimens were placed in the same culture container with their cross-sections facing up and down, creating an environment conducive to decay. The culture containers were placed in a growth chamber at 25 °C and 70% RH for 12 and 24 weeks, respectively. After the decay resistance test, the specimens were removed and placed in an oven at 60 °C for 48 h to determine their mass. The mass loss (ML_d_) values of the specimens were calculated with six repetitions for each group.

### 2.8. Analysis of Variance

All the results were expressed as the mean ± standard deviation (SD). The significance of differences was calculated using Scheffé’s post hoc test following one-way ANOVA, conducted using IBM^®^ SPSS^®^ Statistics for Windows (SPSS, version 20.0). The *p* values < 0.05 were considered to be significant.

## 3. Results and Discussion

### 3.1. Impacts of Different WPGs on the Physical Properties of HTVPA-Modified Japanese Cedar Wood

Wood typically undergoes changes in dimensions due to the swelling of cell walls after the acetylation process. Therefore, to understand the effects of different WPGs on the dimension change (DC) values of HTVPA-modified Japanese cedar wood, we categorised modified specimens into four groups based on their weight gains: WPG 8, WPG 11, WPG 14, and WPG 17. As shown in Table 1, for the modified wood with WPG 8, 11, 14, and 17, the DC values in the tangential direction were 2.9%, 3.6%, 4.4%, and 4.6%, respectively. These DC values increased with increasing WPG values, and a similar trend was observed in the radial direction. Furthermore, as shown in Table 1, except for WPG 17, no statistically significant differences in density between untreated and acetylated wood were observed. The densities of the various specimens ranged from 0.26 to 0.48 g/cm^3^. On the other hand, untreated Japanese cedar wood had an EMC of 11.84%, while the wood treated with HTVPA ranged from 3.28% to 4.02%. These values significantly differed from the untreated wood, and they decreased as the WPG increased. In contrast, the MEE (66.0–72.3%) of HTVPA-modified wood increased with increasing WPG. Additionally, the untreated samples had an S_65_ of 5.1%, while the HTVPA-modified samples showed S_65_ ranging from 0.4% to 1.4%. These results exhibited significant differences between the untreated and HTVPA-modified wood. Additionally, the ASE of HTVPA-modified wood ranged from 73.3% to 92.7%, and this value increased with increasing WPG. These results indicated that Japanese cedar wood after HTVPA treatment demonstrated improved dimensional stability, and the degree of improvement was positively correlated with WPG. This finding was consistent with the impact of acetylation treatment at 120 °C on the dimensional stability of pine wood, as reported by Sun et al. [33].

To further understand the influences of different WPGs on the acetyl group stability in acetylated Japanese cedar wood by the HTVPA process, we conducted five cycles of water-soaking/oven-drying tests and calculated the S_max_ and ASE values of the specimens to assess the acetyl group stability in modified wood with varying WPGs. Figure 2 shows that all the HTVPA-modified specimens had S_max_ values ranging from 1.1% to 4.4%, which were significantly lower than those of untreated specimens (11.8–12.6%). The WPG 17 specimens showed the lowest S_max_ values (1.1% to 1.4%) after five cycles of the water-soaking/oven-drying test. Additionally, the results of the ASE in Table 2 indicated that within the same cycle, the ASE of the acetylated specimens increased with increasing WPGs, ranging from 64.3–65.2% (WPG 8) to 87.5–90.6% (WPG 17). Furthermore, there was little variation in the ASE after five cycles of the water-soaking/oven-drying test for each group. This finding suggested that HTVPA treatment not only effectively enhanced the dimensional stability of wood but also maintained the good dimensional stability of the specimens after repeated water-soaking and oven-drying cycles.

### 3.2. Impacts of Different WPGs on the Termite Resistance of HTVPA-Modified Japanese Cedar Wood

To assess the impacts of different WPGs on the termite resistance of HTVPA-modified wood, indoor termite resistance tests were conducted on untreated and treated wood, evaluating their mass loss (ML_t_) and termite mortality (TM) values. As shown in Table 3, the ML_t_ of untreated wood was 20.3%, while the ML_t_ of HTVPA-modified wood ranged from 6.6% to 3.2%, decreasing with increasing WPG. Furthermore, the TM for untreated wood was 55%, while the TM for HTVPA-modified wood was 95–100%. The main reason for the termite mortality of HTVPA-modified wood was speculated to be related to the replacement of hydroxyl groups in the wood with acetyl groups. This substitution prevented the native microorganisms in the termite digestive system from breaking down these chemically modified components, ultimately depriving the termites of nutrition and causing their death [34]. As shown in Figure 3A, in the termite resistance test, untreated wood still had termite activity after 21 d of testing. In contrast, HTVPA-modified specimens still had termites alive on the 7th day, but when the test reached 14 d, there was no termite activity around the specimens, indicating termite mortality. Additionally, as shown in Figure 3B, untreated specimens had their contours chewed away after the termite resistance test, while HTVPA-modified specimens still maintained certain contours, demonstrating that HTVPA treatment effectively improved the termite resistance of Japanese cedar wood.

Conversely, we compiled the results of termite resistance tests on Japanese cedar wood treated with other common treatments. As shown in Table 3, the ML_t_ and TM values of acetylated wood prepared by the HTVPA process in the present study were 6.6% and 95%, respectively. These values were similar to those of Nagasawa et al. [35] for liquid-phase acetylated wood, indicating that this modification method achieved a similar improvement in termite resistance as liquid-phase modification. Additionally, the HTVPA treatment used in this study reached a temperature of 200 °C, which fell within the range of heat treatment. Relative to Chen et al. [36], who treated Japanese cedar wood at 210 °C for 4 h, it was found that the ML_t_ and TM values of heat-treated wood were 24.7% and 12.7%, respectively, indicating that heat treatment alone did not significantly improve termite resistance in Japanese cedar wood. Furthermore, relative to other common wood preservatives, such as alkaline copper quaternary (ACQ) and copper azole (CuAz) [37], the ML_t_ value of Japanese cedar wood modified by the HTVPA process was slightly increased, but its TM value significantly improved.

### 3.3. Impacts of Different WPGs on the Decay Resistance of HTVPA-Modified Japanese Cedar Wood

To assess the impacts of different WPGs on the decay resistance of modified Japanese cedar wood, we followed the performance standard CNS 6717 [32] for wood preservatives. Decay resistance tests were conducted for 12 and 24 weeks using brown-rot and white-rot fungi on both untreated and HTVPA-modified Japanese cedar wood. As shown in Table 4, after 12 weeks of decay resistance testing by brown-rot fungi (LS and FP), untreated wood exhibited mass loss (ML_d_) values of 39.6% and 54.5%, respectively. In contrast, modified wood showed ML_d_ values of 0.2–0.9% and −0.2–0.3%, respectively, with no significant differences observed among the modified samples. When the testing period was extended to 24 weeks, the ML_d_ values for untreated wood in the LS and FP tests increased to 48.5% and 57.9%, respectively, while HTVPA-modified wood exhibited ML_d_ values of 2.4–2.8% and −0.5–−0.1%, respectively, with no significant differences among the modified samples. These results indicated that HTVPA treatment could improve the decay resistance of Japanese cedar wood against brown-rot fungi. Conversely, after 12 weeks of decay resistance testing by white-rot fungi (LB and TV), untreated wood displayed ML_d_ values of 16.3% and 39.1%, respectively, while HTVPA-modified wood exhibited ML_d_ values of 0.3% and 0.1–0.2%, respectively. Similarly, when the testing period was extended to 24 weeks, untreated wood showed ML_d_ values of 19.7% and 39.0% in the LB and TV tests, respectively, while modified wood had ML_d_ values of 0.5–0.9% and 0.0–0.2%, respectively. These results demonstrated that HTVPA treatment could enhance the decay resistance of Japanese cedar wood against white-rot fungi. Generally, brown-rot fungi are more harmful to softwood than white-rot fungi. Hoseeinpourpia and Mai [38] noted that the decay mechanism of wood-decaying fungi could be divided into two stages. The first stage involved the chelator-mediated Fenton system (CMF system), where the chelator served as a mediator. Through the Fenton reaction (Fe^2+^ + H_2_O_2_ + H^+^ → Fe^3+^ + H_2_O + ^•^OH), hydroxyl radicals (^•^OH) formed, which then reacted with the cellulose, hemicellulose, and lignin in the wood cell walls. This phenomenon resulted in the degradation of the wood cell walls and the formation of voids. Simultaneously, phenolic derivatives produced by brown-rot fungi could reduce Fe^3+^ in the microenvironment to Fe^2+^, allowing the degradation reaction to continue uninterrupted. The second stage was the enzyme system, which primarily utilised the voids in the cell wall to allow carbohydrate-degrading enzymes to enter and react. Wood acetylation could affect CMF degradation by lowering the capacity for iron absorption, likely due to the loss of OH groups [39]. This result made it difficult for the Fenton reaction to occur, thereby achieving decay resistance.

**Table 3 polymers-16-01601-t003:** Mass loss (ML_t_) and termite mortality (TM) values of untreated, acetylated, and treated Japanese cedar wood after the termite resistance test.

Treatment ^†^	ML_t_ (%)	TM (%)	Reference
Untreated	20.3 ^A^ ± 4.7	55 ^B^ ± 16	Present study ^††^
HTVPA (WPG 11)	6.6 ^B^ ± 0.3	95 ^A^ ± 5	
HTVPA (WPG 14)	7.3 ^B^ ± 0.4	99 ^A^ ± 2	
HTVPA (WPG 17)	3.2 ^B^ ± 0.8	100 ^A^ ± 0	
ACQ	0.87 ± 0.15	50.8 ± 4.6	[37]
CuAz	0.89 ± 0.18	63.8 ± 7.8	
210 °C/4 h	24.7 ± 6.2	12.7 ± 3.1	[36]
LPA (WPG 22.8)	5	89	[35]

^†^ ACQ: alkaline copper quaternary; CuAz: copper azole; 210 °C/4 h: heat treatment at 210 °C for 4 h; LPA: liquid-phase acetylation. The copper retention levels of the ACQ and CuAz samples was 1.61 ± 0.12 and 1.28 ± 0.12 kg/m^3^, respectively. ^††^ Values are mean ± SD (*n* = 3). Different capital letters within a column indicate significant differences among various WPGs by one-way ANOVA and the Scheffé test (*p* < 0.05).

In Figure 4A,B, the mycelial growth patterns of untreated and HTVPA-modified wood after 12 and 24 weeks of decay resistance testing could be observed. It was evident that untreated wood was covered with mycelium on its surface after decay testing. In contrast, HTVPA-modified wood exhibited a reduced mycelial distribution, and there was less white-rot than brown-rot. The primary reasons for this phenomenon were as follows: (1) acetylation treatment reduced the moisture content of the wood, which could slow the Fenton reaction of decay fungi; (2) white-rot fungi required higher levels of moisture for growth than brown-rot fungi. Therefore, white-rot fungi tended to grow slowly in environments with insufficient moisture. In addition, from the images of test specimens in Figure 4C,D after 12 and 24 weeks of decay resistance testing, we observed that untreated wood underwent severe deformation after decay by FP fungi. After decay by TV fungi, untreated wood exhibited colour fading. In contrast, HTVPA-modified wood maintained its original colour and shape after decay by both FP and TV fungi. This finding demonstrated that HTVPA treatment effectively enhanced the decay resistance of Japanese cedar wood. The main reason behind this phenomenon could be attributed to the fact that brown-rot fungi primarily degraded cellulose and to a lesser extent lignin, whereas white-rot fungi mainly targeted lignin. Consequently, untreated wood tended to deform after decay by brown-rot fungi, while it exhibited colour fading after decay by white-rot fungi.

Conversely, we compared HTVPA treatment with various common wood preservations, such as chromated copper arsenate (CCA), ACQ, boron didecyldimethylammonium chloride (BAAC), CuAz, cyproconazole/imidacloprid (AZN), and alkyl ammonium compounds (AACs) [40,41], and the results are presented in Table 5. Accordingly, we observed that the decay resistance of the HTVPA-modified specimens with WPG 8 was similar to that of wood preservatives. Furthermore, relative to single treatment methods, such as heat treatment at 210 °C for 4 h [36], liquid-phase acetylation [18], low-temperature vapour-phase acetylation [18], and supercritical carbon dioxide acetylation [42], HTVPA-modified wood exhibited lower ML_d_. This result suggested that the HTVPA treatment proposed in this study effectively enhanced the decay resistance of wood. This treatment not only achieved a similar improvement to commercially available wood preservatives, but also outperformed single heat treatment or acetylation methods. Additionally, according to the performance standard CNS 6717 [32] for wood preservatives, wood with an ML_d_ value lower than 3% was considered to have good decay resistance. The acetylated Japanese cedar wood prepared in this study all showed ML_d_ values below 3%, indicating excellent decay resistance against LS, FP, LB, and TV.

## 4. Conclusions

We primarily utilised the HTVPA process, which is a simultaneous acetylation and heat treatment modification, to prepare Japanese cedar wood with varying WPGs. The aim was not only to investigate their physical properties but also to evaluate their termite and decay resistance. The results showed that the EMC and S_65_ of HTVPA-modified wood decreased with increasing WPG. Conversely, the DC, MEE, and ASE values increased with increasing WPG. Even after five cycles of water-soaking and oven-drying tests, the ASE did not change significantly, indicating that the Japanese cedar wood modified through HTVPA treatment demonstrated excellent dimensional stability. In addition, the lab-scale termite resistance test indicated that the ML_t_ and TM of untreated wood were 20.3% and 55%, respectively, while those of modified wood were 3.2–6.6% and 95–100%, respectively. This result indicated a significant improvement in the termite resistance of the wood after HTVPA treatment.

Furthermore, the results of a 24-week lab-scale decay resistance test revealed that the ML_d_ values of untreated wood for LS, FP, LB, and TV were 48.5%, 57.9%, 19.7%, and 39.0%, respectively. In contrast, modified wood had ML_d_ values of 2.4–2.8%, −0.5–−0.1%, 0.5–0.9%, and 0.0–0.2%, respectively. There were no significant differences among the HTVPA-modified wood with varying WPGs, and all met the standards for the ML_d_ values of decay-resistant materials in CNS 6717 (below 3%). Additionally, the results of this study indicated that HTVPA treatment was more effective in enhancing wood decay resistance than separate heat treatment or conventional liquid-phase acetylation. However, both the lab-scale termite resistance test and 24-week decay resistance test do not fully represent actual outdoor conditions. Therefore, further investigation is needed to evaluate termite and fungal damage in outdoor environments. Based on these findings, subsequent research could further focus on the weathering properties and actual biotic resistance of HTVPA-treated wood in outdoor environments to assess its potential for outdoor applications.

## Figures and Tables

**Figure 1 polymers-16-01601-f001:**
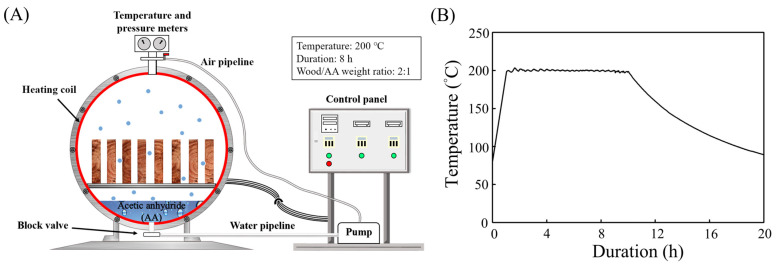
Schematic diagram (**A**) and heating curve (**B**) of the HTVPA process.

**Figure 2 polymers-16-01601-f002:**
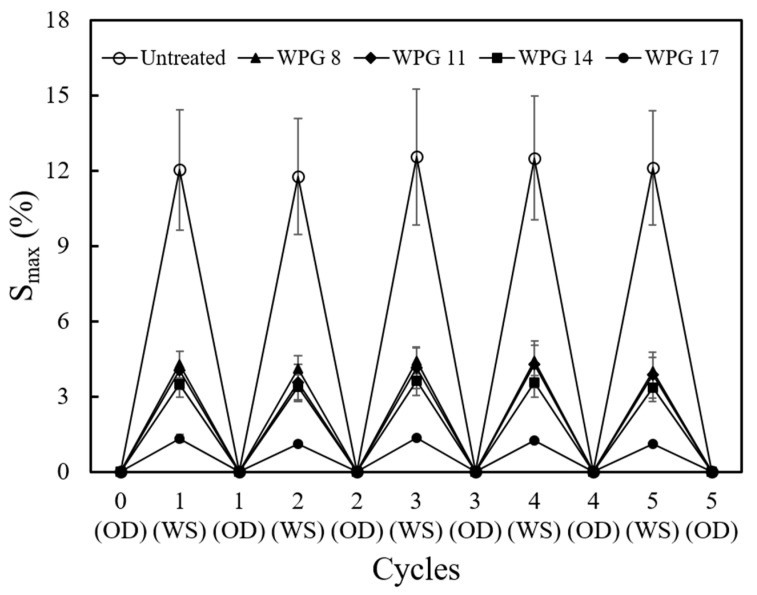
Total volumetric swelling (S_max_) of untreated and HTVPA-modified Japanese cedar wood samples with different cycles of water-soaking and oven-drying tests. OD: oven-drying. WS: water soaking. Values are mean ± SD (*n* = 7).

**Figure 3 polymers-16-01601-f003:**
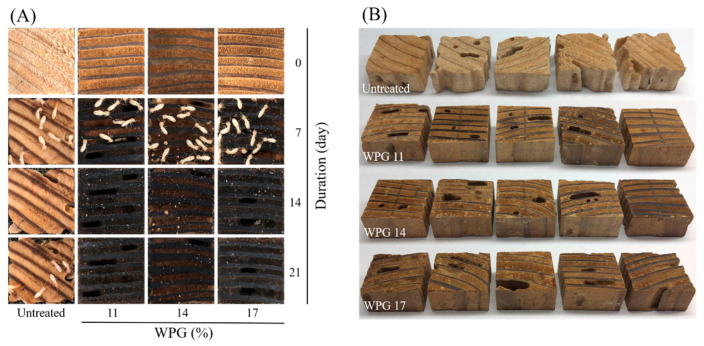
Images of untreated and HTVPA-modified Japanese cedar wood during (**A**) and after (**B**) the termite resistance test.

**Figure 4 polymers-16-01601-f004:**
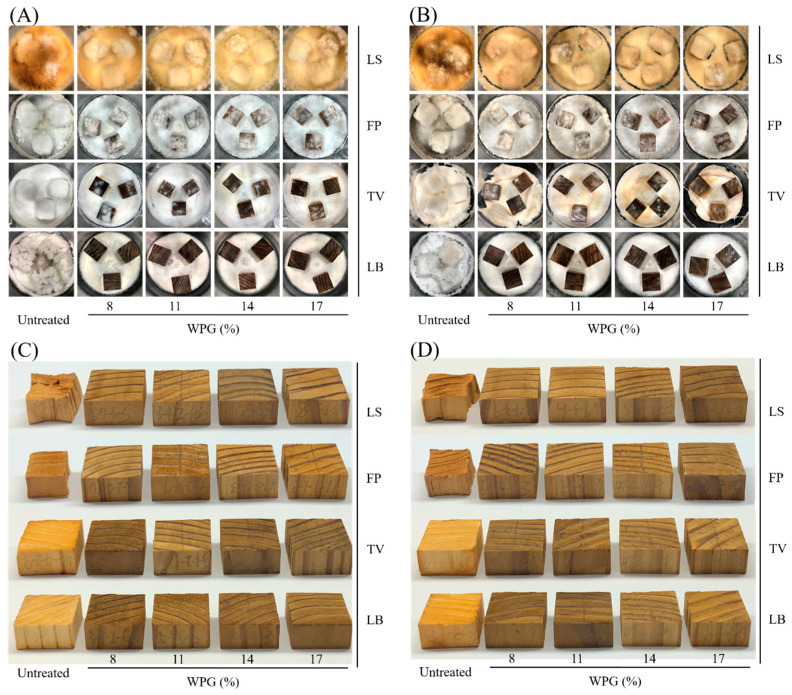
Mycelial growth and images of untreated and HTVPA-modified Japanese cedar wood after 12 (**A**,**C**) and 24 (**B**,**D**) weeks of the decay resistance test.

**Table 1 polymers-16-01601-t001:** Weight percent gain (WPG), dimensional change (DC), density, equilibrium moisture content (EMC), moisture excluding efficiency (MEE), volumetric swelling (S_65_), and anti-swelling efficiency (ASE_65_) of Japanese cedar wood before and after HTVPA treatment.

Sample Code	WPG (%)	DC (%)		Density (g/cm^3^)	EMC (%)	MEE (%)	S_65_ (%)	ASE_65_ (%)
		Tangential	Radial					
Untreated	–	–	–	0.42 ^A^ ± 0.08	11.84 ^A^ ± 0.07	–	5.1 ^A^ ± 1.4	–
WPG 8	7.0–9.0	2.9 ^C^ ± 0.1	1.8 ^B^ ± 0.1	0.48 ^A^ ± 0.05	4.02 ^B^ ± 0.26	66.0 ^B^ ± 2.2	1.3 ^B^ ± 0.1	74.6 ^B^ ± 2.2
WPG 11	10.0–12.0	3.6 ^B^ ± 0.1	2.5 ^A^ ± 0.1	0.45 ^A^ ± 0.07	4.02 ^B^ ± 0.43	66.0 ^B^ ± 3.7	1.4 ^B^ ± 0.3	73.3 ^B^ ± 5.0
WPG 14	13.0–15.0	4.4 ^A^ ± 0.2	2.6 ^A^ ± 0.3	0.44 ^A^ ± 0.06	3.82 ^B^ ± 0.35	67.8 ^B^ ± 3.0	1.2 ^B^ ± 0.2	77.2 ^B^ ± 4.5
WPG 17	16.0–18.0	4.6 ^A^ ± 0.4	2.5 ^A^ ± 0.2	0.26 ^B^ ± 0.004	3.28 ^C^ ± 0.08	72.3 ^A^ ± 0.7	0.4 ^C^ ± 0.1	92.7 ^A^ ± 1.3

Values are mean ± SD (*n* = 7). Different capital letters within a column indicate significant differences among various WPGs by one-way ANOVA and the Scheffé test (*p* < 0.05).

**Table 2 polymers-16-01601-t002:** Anti-swelling efficiency (ASE) of HTVPA-modified Japanese cedar wood with different cycles of water-soaking and oven-drying tests.

Sample Code	ASE (%)				
	Cycle 1	Cycle 2	Cycle 3	Cycle 4	Cycle 5
WPG 8	64.4 ^Ba^ ± 4.5	65.2 ^Ba^ ± 4.4	64.8 ^Ba^ ± 4.1	64.3 ^Ba^ ± 4.8	64.6 ^Ba^ ± 4.5
WPG 11	66.3 ^Ba^ ± 6.5	70.1 ^Ba^ ± 6.2	66.9 ^Ba^ ± 6.5	65.5 ^Ba^ ± 7.4	65.7 ^Ba^ ± 7.3
WPG 14	71.0 ^Ba^ ± 4.1	71.6 ^Ba^ ± 4.2	71.2 ^Ba^ ± 4.5	71.2 ^Ba^ ± 4.7	69.7 ^Ba^ ± 4.5
WPG 17	89.0 ^Ab^ ± 1.3	90.6 ^Aa^ ± 0.8	89.2 ^Aab^ ± 0.5	89.8 ^Aab^ ± 0.8	87.5 ^Ac^ ± 0.5

Values are mean ± SD (*n* = 7). Different capital letters within a column indicate significant differences among various WPGs by one-way ANOVA and the Scheffé test (*p* < 0.05). Different lowercase letters within a row indicate significant differences among various cycles by one-way ANOVA and the Scheffé test (*p* < 0.05).

**Table 4 polymers-16-01601-t004:** Mass loss (ML_d_) values of untreated and HTVPA-modified Japanese cedar wood after 12 and 24 weeks of the decay resistance test.

Sample Code	Periods (Weeks)	Brown Rot		White Rot	
		LS	FP	LB	TV
Untreated	12	39.6 ^A^ ± 13.4	54.5 ^A^ ± 4.3	16.3 ^A^ ± 8.6	39.1 ^A^ ± 3.1
WPG 8		0.9 ^B^ ± 0.7	0.3 ^B^ ± 0.5	0.3 ^B^ ± 0.1	0.1 ^B^ ± 0.1
WPG 11		0.7 ^B^ ± 0.6	−0.2 ^B^ ± 0.2	0.3 ^B^ ± 0.1	0.2 ^B^ ± 0.2
WPG 14		0.2 ^B^ ± 0.2	0.0 ^B^ ± 0.2	0.3 ^B^ ± 0.2	0.1 ^B^ ± 0.2
WPG 17		0.3 ^B^ ± 0.4	−0.2 ^B^ ± 0.2	0.3 ^B^ ± 0.1	0.1 ^B^ ± 0.5
Untreated	24	48.5 ^A^ ± 2.7	57.9 ^A^ ± 2.2	19.7 ^A^ ± 14.7	39.0 ^A^ ± 7.6
WPG 8		2.4 ^B^ ± 0.6	−0.3 ^B^ ± 0.7	0.7 ^B^ ± 0.2	0.1 ^B^ ± 0.2
WPG 11		2.8 ^B^ ± 0.6	−0.2 ^B^ ± 0.3	0.9 ^B^ ± 0.1	0.2 ^B^ ± 0.5
WPG 14		2.8 ^B^ ± 0.6	−0.1 ^B^ ± 0.6	0.6 ^B^ ± 0.2	0.2 ^B^ ± 0.3
WPG 17		2.6 ^B^ ± 1.6	−0.5 ^B^ ± 0.6	0.5 ^B^ ± 0.3	0.0 ^B^ ± 0.2

Values are mean ± SD (*n* = 6). Different capital letters within a column indicate significant differences among various WPGs by one-way ANOVA and the Scheffé test (*p* < 0.05).

**Table 5 polymers-16-01601-t005:** Mass loss (ML_d_) values of treated Japanese cedar wood after 12 weeks of the decay resistance test.

Treatment ^†^	ML_d_ (%)		Reference
	Brown Rot (FP)	White Rot (TV)	
Untreated	54.5 ± 4.3	39.1 ± 3.1	Present study (adapted from Table 4)
HTVPA (WPG 8)	0.3 ± 0.5	0.1 ± 0.1	
HTVPA (WPG 11)	−0.2 ± 0.2	0.2 ± 0.2	
HTVPA (WPG 14)	0.0 ± 0.2	0.1 ± 0.2	
HTVPA (WPG 17)	−0.2 ± 0.2	0.1 ± 0.5	
CCA (WPG 12.8)	0.25	–	[40]
ACQ (WPG 7–8)	2	–	
BAAC	−1	2	[41]
CuAz	−1	2	
AZN	0	2	
AAC	1	2	
210 °C/4 h	0.75 ± 0.08	0.95 ± 0.13	[36]
VPA (WPG 16)	−0.02 ± 0.11	0.33 ± 0.09	[18]
LPA (WPG 20)	−0.44 ± 0.19	0.28 ± 1.11	
CO_2_A (WPG 22)	1	–	[42]

^†^ CCA: chromated copper arsenate. ACQ: alkaline copper quaternary. BAAC: boron didecyldimethylammonium chloride. CuAz: copper azole. AZN: cyproconazole/imidacloprid. AAC: alkyl ammonium compounds. 210 °C/4 h: heat treatment at 210 °C for 4 h. VPA: vapour-phase acetylation. LPA: liquid-phase acetylation. CO_2_A: acetylation by supercritical carbon dioxide.

## Data Availability

The original contributions presented in the study are included in the article, further inquiries can be directed to the corresponding authors.

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
