# Peer review of "Decay and Termite Resistance of Wood Modified by High-Temperature Vapour-Phase Acetylation (HTVPA), a Simultaneous Acetylation and Heat Treatment Modification Process"

_polymers, 2024, doi:10.3390/polym16111601_

Round 1
Reviewer 1 Report
Comments and Suggestions for Authors
Manuscript "Decay and Termite Resistance of Wood Modified by High-Temperature Vapour-Phase Acetylation (HTVPA), a Simultaneous Acetylation and Heat Treatment Modification Process" is very interesting.
General comments:
The aim of this study was to investigate the decay and termite resistance of Japanese cedar wood modified by HTVPA. Authors analysed impact of HTVPA treatment on the resistance of wood to biological degradation. Authors exhibited a mass loss of 20.3%, while HTVPA-modified wood showed a reduced MLt of 6.6–3.2%, which decreased with an increase in weight percent gain, and the termite mortality reached 95–100%.
Detailed comments:
The Introduction is interesting and correctly written.
The study material is also well described. Unfortunately, the description of the methodology is not complete. In particular, the description of the statistical methods used in the study is very poor. And in fact there is none. The title of the subsection "2.8 Analysis of variance" does not describe the analysis of variance. All the methods used in the research should be described here so that anyone can repeat the experience. Unfortunately, the manuscript does not include these descriptions. Were assumptions tested to allow for the use of analysis of variance? By what method?
Table 1: Letters indicate homogeneous groups. They refer to mean values, so the letters should be placed directly after the mean values, not after the SD.
Table 1 should be completed with the critical values that were used to construct homogeneous groups.
Table 2: Letters indicate homogeneous groups. They refer to mean values, so the letters should be placed directly after the mean values, not after the SD.
Table 2 should be completed with the critical values that were used to construct homogeneous groups.
Table 3: Letters indicate homogeneous groups. They refer to mean values, so the letters should be placed directly after the mean values, not after the SD.
Table 3 should be completed with the critical values that were used to construct homogeneous groups.
Table 4 is after Table 5!
Table 4: Letters indicate homogeneous groups. They refer to mean values, so the letters should be placed directly after the mean values, not after the SD.
Table 4 should be completed with the critical values that were used to construct homogeneous groups.
The results shown in Table 4 suggest that the experiment was two-way, not one-way, as the Authors wrote. The analysis should be re-run and the results of the interaction of the two factors should be reported.
The authors analyze many features. Unfortunately, they did not even conduct a correlation analysis. Why? The results of such an analysis would have been an interesting addition to the results obtained.
Paper needs major revision.
Reviewer 2 Report
Comments and Suggestions for Authors
Title: Decay and Termite Resistance of Wood Modified by High-Temperature Vapour-Phase Acetylation (HTVPA), a Simultane ous Acetylation and Heat Treatment Modification Process
Overall Comments:
The paper uses acetic anhydride to describe the modified wood's resistance to termite attacks and decay using high-temperature vapor-phase acetylation (HTVPA). The authors published the HTVPA process in a journal and studied only the above-mentioned test. The acetic anhydride used for this study is a banned chemical in various countries. Additionally, the mechanical properties of the treated samples were not analyzed. The presentation of graphs and tables is not acceptable. However, the paper is interesting, and improvement is needed for publication in this journal.
Comments on the Title, Abstract, and References:
1. Authors may consider revising the title to "Decay and Termite Resistance of Wood Modified by High-Temperature Vapour-Phase Acetylation."
2. The abstract is acceptable.
Comments on the Introduction:
3. Lines 48 to 57: The authors should adjust these lines to form a concluding paragraph that explains the objective and novelty of this study.
4. Lines 62-63: Elaborate on the previous modification techniques and their advantages and disadvantages.
5. Line 67: "Acetic anhydride or other chemicals to form acetyl groups." List the names of other chemicals.
6. Please avoid bulk references like [20-26]. The authors should discuss the key findings of each cited reference.
Comments on the Experiment:
7. The materials section needs to be modified. It appears to be copied and pasted from the authors' previous article.
8. Please include the names of the chemicals used for this study in the materials section.
9. Please include sample details in a table, including their size, weight, moisture content, dimensions, thickness, volume, and density.
10. Figure 1A needs modification. Otherwise, the authors should mention the copyright permission from their previous publisher in the caption for Figure 1.
11. Figure 1A should be in this section, not the results and discussion.
12. What is AA? Is it acetic acid or acetic anhydride?
13. How many samples have been used for this test, and what other tests should be included?
14. If it is acetic anhydride, it has been banned in various countries for the illicit manufacture of heroin and methaqualone. Besides, it is expensive.
15. Line 116: "The specimens were suspended on an iron rack to prevent direct contact with AA." What does "suspended" mean in this context?
16. Include the water used for this experiment.
17. The authors should rewrite the HTVPA process in detail here to enable the reproduction of this work.
18. It is suggested that the authors include the EMC and MEE analysis procedures.
Comments on Results and Discussions:
19. How did the color of the samples change after the termite resistance test?
20. Visually, it is obvious that termites effectively chewed the untreated wood and then in WPG 17, WPG 14, and WPG 11. What are the reasons for this?
21. What are the results for WPG 8 samples?
22. Lines 274-275: "In contrast, HTVPA-modified specimens still had termites alive on the 7th day, but when the test reached 14 days, there was no termite activity around the specimens, indicating termite mortality." Did this happen for all the treated samples?
23. What will these treated samples' mechanical properties (hardness, MOE, MOR)?
Comments on Conclusions:
24. Lines 405-409: Check the meaning and clarify "or conventional liquid-phase acetylation."
25. The reviewer suggests rewriting the conclusion section to include the limitations and utility of this study.
26. Please check by iThenticate before submitting the revised version. Results should be below 20%.
Round 2
Reviewer 1 Report
Comments and Suggestions for Authors
The authors revised the manuscript according to all my comments and suggestions. In its present form, it can be published.
Reviewer 2 Report
Comments and Suggestions for Authors
Dear Authors,
Thank you for improving your manuscript according to the reviewers' comments. I am now recommending it for publication.
Best wishes!